

# Assessing shadow effects on Photochemical Reflectance Index (PRI) for the water stress detection in winter wheat

Xin Yang[1], Shishi Liu[1]*, Yinuo Liu[1], Xifeng Ren[2], Hang Su[1]

[1]School of Resources and Environment, Huazhong Agricultural University, Wuhan, 430070, China
[2]College of Plant Science and Technology, Huazhong Agricultural University, Wuhan, 430070, China

*Correspondence to*: Shishi Liu (carol.shishi@gmail.com)

**Abstract.** The photochemical reflectance index (PRI) has emerged to be a pre-visual indicator of water stress. However, whether the varying shadow fraction, which may be caused by multiple view angles or the changing crop density in the field, affects the performance of PRI in detecting water stress of crops is still uncertain. This study evaluated the impact of the

varying shadow fraction on estimating relative water content (RWC) across growth stages of winter wheat using different formulations of PRI. Results demonstrated that PRI570, PRI1, and PRI2 of shadow were higher than those of sunlit leaves for unstressed plants, but the contrary results were achieved for stressed plants. Despite the difference between PRI_shadow and PRI_leaf, the significance of the linear relationship between RWC and PRI did not change with the different ratio of sunlit leaves and shadow. For most studied PRI formulations, the slope and intercept of the linear regression model between

PRI and RWC changed proportionally with the shadow fractions. We applied a uniform RWC prediction model to the data of varying shadow fractions and found that the accuracy of RWC predictions was not significantly affected, indicating that the effect of varying shadow fractions was minimal to the seasonal water stress detection in winter wheat using PRI.

## 1 Introduction

Agriculture consumes about 80%-90% of fresh water worldwide (Gonzalez-Dugo, Durand, and Gastal, 2010). Water

stress is one of the most critical abiotic stressors limiting plant growth and crop production (Chaves, Maroco, and Pereira, 2003). Climate change, increasing worldwide shortages of water, frequent droughts are exacerbating the agricultural water crisis and putting global food security at risk (Hirich et al. 2016; Lei et al., 2016). The assessment of water status in crops is critical for precision irrigation practices, balancing crop production with the water supply and sustainable farming. Remote sensing provides a unique tool to unobtrusively, efficiently and quantitatively assessing water status in crops. As water stress

induces plants' stomatal closure, leading to the increasing leaf temperature due to the decreasing evaporative cooling, remotely monitoring the change in canopy temperature is a direct way to evaluate water stress. Monitoring water stress with thermal remote sensing has been accomplished using spectrometers at ground level (Idso et al., 1981), thermal sensors at image level , and satellite thermal information at large scales (Sayago, Ovando, and Bocco., 2017). However, the thermal remote sensing of water stress has limitations in both physiological and operational aspects. The physiological relationship



between canopy temperature and stress is not clear for some crops (Villalobos, Testi, and Moreno-Perez, 2009). Due to the technical reasons, the spatial resolution of thermal imaging sensors is generally coarser than the visible and infrared sensors, limiting its applications at local scales.

In a recent decade, the photochemical reflectance index (PRI) has emerged to be a pre-visual indicator of water stress. PRI is a normalized difference of reflectance at 531 nm and reflectance at a reference band (e.g. 570 nm) in the visible domain. It is initially proposed as an indicator of the de-epoxidation state of the xanthophyll pigments, relating to photosynthesis (Gamon, Peñuelas, and Field, 1992). When the light absorbed by the plants exceeds the photosynthetic demand, de-epoxidation of xanthophyll cycle pigments occurs and results in the downregulation of photosynthesis (Gamon, Peñuelas, and Field, 1992). Water stress is one of the important triggers to the xanthophyll cycle, leading to the apparent drop in reflectance at 531 nm (Muller 2001; Sun et al., 2008; Sarlikioti, Driever, and Marcelis, 2010; Zarco-Tejada et al., 2013). Several previous studies investigate the feasibility of using PRI to assess plant water status at leaf level and canopy level. At leaf level, a number of studies demonstrate a close relationship between PRI and physiological indicators of water stress (Thenot, Méthy, and Winkel, 2002; Shahenshah et al., 2010), but some other studies report a poorer relationship due to the confounding environmental factors (Sarlikioti, Driever, and Marcelis, 2010) or the changes in pigment pools (Sun et al., 2008). At canopy level, the ability of PRI for water stress detection is affected by canopy structure, canopy cover and viewing geometry (Rossini et al. 2013; Panigada et al., 2014). Particularly, at seasonal and inter-annual time scales, physiological changes, such as relative water content and pigment pools, concurrently occur with structural changes, such as leaf area index (LAI). The limited long-term studies show that canopy PRI is sensitive to the structural changes during the growth season (Gitelson, Gamon, and Solovchenko, 2017), which significantly affects its capabilities to detect water stress (Suárez et al., 2010; Zarco-Tejada et al., 2013). To minimize the impact of canopy structures on PRI, transformations of PRI are developed using the band insensitive to the canopy structure (Hernández-Clemente et al., 2011), the structural vegetation indices (VIs) for the normalization (Zarco-Tejada et al., 2013; Gitelson, Gamon, and Solovchenko, 2017), or the radiative transfer modeling results (Hernández-Clemente et al., 2011).

While improvements have been achieved for detecting water stress with canopy PRI, the impacts of shadowing on PRI and its capability to detect water stress are generally ignored. PRI is primarily driven by the xanthophyll cycle at the short time scale (e.g. a few hours, two to three days), but shaded leaves may not experience de-epoxidation of xanthophyll cycle as the sunlit leaves do. As PRI is expected to be applied to monitoring water stress at large scale, canopy PRI derived from satellite data includes contributions from both the sunlit leaves and shaded leaves. Hall et al. (2008) and Hilker et al. (2010) found that canopy PRI is strongly dependent on canopy shadow fractions, and that the directional changes observed in PRI at a given 15 minutes or half hour interval can be attributed to changes in canopy shadow fractions. Cheng et al. (2009) examined the contributions of variable sunlit/shaded canopy ratios to the simulation of canopy PRI with the two-layer Markov chain analytical canopy reflectance model, confirming the importance of adding shaded leave in the simulation. Takala and Mõttus (2016) demonstrated that the illumination-induced shadowing effects explained the observed dynamic range of apparent canopy PRI derived from the high spatial resolution airborne imaging spectroscopy data.



Previous studies have shown that within-canopy shadowing effects directly affect PRI of a canopy, but whether the shadowing effect further influences the PRI's capability in detecting water stress in the growth season of a crop is still uncertain. The objective of this study is to analyze the impact of varying shadow fractions on the performance of canopy PRI in detecting water stress during the growth season of winter wheat using a hyperspectral imager. To accomplish this objective, we conducted water stress experiments of winter wheat for two consecutive years. Reflectance of shaded and sunlit leaves derived from hyperspectral imagery were mixed with varying fractions to quantify the shadow effects on different formulations of PRI in detecting water stress at the seasonal scale.

## 2 Materials and Methods

### 2.1. Study Site and Experimental Design

During the growth seasons of 2016 and 2017, two water stress experiments were conducted in the facilities at Huazhong Agricultural University, China (30°28′N, 114°22′E). The mean annual temperature is approximately17.0 °C and the mean annual total precipitation is around 1256 mm. The seeds of cultivar 'Zheng 9023', which is widely planted in central China, were used in the experiment. Seeds were sown on November 2nd, 2015 and November 26th, 2016 respectively, in a rectangle plastic pot (L70cm×W40cm×H35cm) with the density of approximately 250-300 seeds/pot. The soil was silt loam, with the volumetric water content of 26% at the field capacity. Sufficient NPK (5:4:1) fertilizer was applied in the soil before sowing. The experiments consisted of 28 pots in 2015-2016 and 15 plots in 2016-2017. Pest and disease control were conducted in the same time during the growth period, to ensure the plants were not under additional stresses other than different levels of water stress.

Seedlings were growing outdoor under the natural condition before the water stress experiments started. Soil water content was measured every 4-5 days using a time domain reflectometry (TDR300, Spectrum Technology Inc., USA), and tap water was supplied if soil water content was below 70% of the field capacity. The water stress treatments started at the end of February, which was the tiller initiation stage. Pots were moved to a rain-out shelter to prevent the external water supply. In 2015-2016, 28 pots were divided into five groups. A group of four pots was used as the reference, which had sufficient water supplies throughout the experiment. The other four groups (with six pots for each group) stopped watering on Feb 24th, March 6th, March 28th, and April 8th respectively. In 2016-2017, 15 pots were divided into six groups. A group of three pots was used as the reference, which had sufficient water supplies throughout the experiment. The other four groups (with three pots for each group) stopped irrigation on March 15th, March 22nd, March 29th, and April 12th respectively. After irrigation stopped, soils of the treated pots were left to dry to analog the natural drought condition. In 2016, measurements were taken every two to five days depending on the weather conditions until immature senescence occurred. For the water treatment group, three pots of winter wheat were used for capturing hyperspectral images, and the other three pots were used to collect samples. In 2017, measurements were taken every four to six days until immature senescence occurred. For the water treatment groups, one pot of winter wheat was used for capturing hyperspectral images, and the other





two pots were used to collect samples. In both years, measurements were taken in control groups during the whole experiment.

## 2.2 Physiological measurements

In this study, we used relative water content (RWC) as the indicator of water stress, because RWC was recommended by previous studies as an effective physiological indicator of water status (Hewitt et al., 1985; Siddique, Hamid, and Islam, 2000). We randomly chose three plants in the sampled pot, and all leaves of the sampled plants were cut from the stem. Leaves were taken into ten small round pieces with a puncher and put into a zip lock bag. Leaf samples were enclosed in a cooler and brought to the laboratory to measure RWC. In the laboratory, fresh weight was measured with an electronic balance. The leaf samples were immersed in distilled water for 16-18 hours. We dried the surface moisture and weighed the turgid weight. Afterward, all samples were put into aluminum boxes to dry in the oven at 105 °C for 15-20 minutes, and then dried at 80 °C for about 10 hours when a constant dry weight was reached. The RWC of leaf samples was calculated as:

$$RWC = \frac{WF - WD}{WT - WD} \quad (\%) \tag{1}$$

where WF is the fresh weight, WT is the turgid weight, and WD is the dry weight.

## 2.3 Spectral data

### 2.3.1 Hyperspectral image acquisition

Hyperspectral images were recorded in situ using SOC710VP Portable Hyperspectral Imager (Surface Optics Corporation, USA). The imager has a spatial resolution of 640×640 pixels and 128 bands in the range of 379-1039nm, with a spectral resolution of 4.6875 nm and a 25° field of view (IFOV). The transparent shed was open half an hour before measuring began. The imager was set up with a nadir view angle and approximately 1.5 m above the canopy. Hyperspectral images were recorded under sunny and cloudless conditions around midday (10:00-14:00) local time. A reference spectral panel was placed on the pot for each measurement to correct for radiation measurement errors due to differences in solar illumination. The spectral data was acquired by LuCamSoftware Camera Drivers and the HyperScanner Software platform. After the hyperspectral images acquisition was completed, the radiometric calibration and wavelength correction were performed using the SOC's Spectral Radiance Analysis Toolkit (SRAnal), converting the raw DN values of the hyperspectral image to reflectance.

### 2.3.2 Hyperspectral image classification

In this study, the Mahalanobis distance method, a supervised classification method was used to classify each hyperspectral image into: leaves, pots, reference plates, ground, and shadows. The supervised classification was performed in ENVI (The Environment for Visualizing Images). The average overall accuracy of all classified images is 0.90, and the Kappa coefficient is 0.83. An example of the original hyperspectral image versus the classified image is shown in Figure 1.





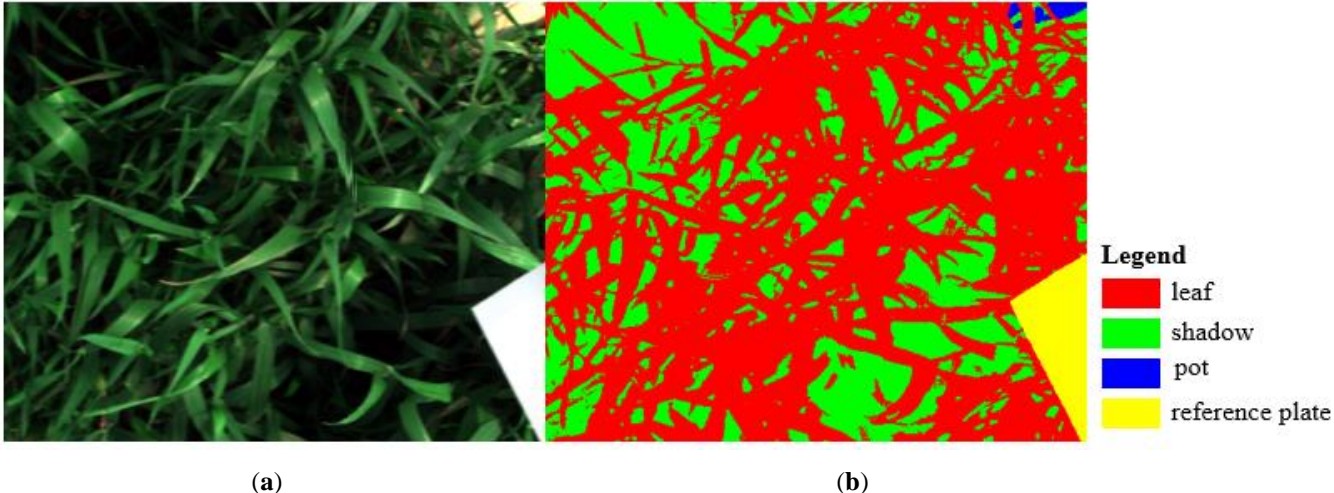

(**a**)                                    (**b**)

**Figure 1.** The original hyperspectral image shown in RGB form (**a**), and the classified image using the Mahalanobis distance classification method (**b**).

### 2.3.3 Spectral reflectance extraction and preprocessing

A region of interest (ROI) of 100×200 pixels was established in the center of each image. Within a ROI, reflectance of pixels identified as green leaves was averaged and used as reflectance of sunlit leaves; reflectance of pixels identified as shadow was averaged and used as reflectance of shadow. Based on the assumption of the linear mixture of shadow and sunlit leaves, we mixed different fractions of shadow reflectance with sunlit-leaf reflectance to evaluate the shadow effect on detecting water stress with PRI. The fraction of shadow and the fraction of sunlit leaves should be summed to 1. We also calculated the average reflectance within the ROI including all the shadow pixels and green-leaf pixels.

The derived spectral data was interpolated to 1 nm band width using the cubic spline interpolation function in the MATLAB software. Twelve existing VIs, including the normalized vegetation index (NDVI), the water index (WI), the ratio of WI and NDVI (WI/NDVI), the corrected red edge normalized vegetation index (mNDVI705), different formulations of PRI (Table 1). In addition, we calculated the difference between VIs of sunlit leaves and VIs of shadow (Equation (2)). In order to provide a more intuitive description, we normalized ΔVI using the maximum and minimum ΔVI. The normalized ΔVI (Equation (3)) was applied to quantify the variations of ΔVI during water stress.

$$\Delta VI = VI\_leaf\_ - VI\_shadow \tag{2}$$

$$\Delta VI\_normalized = \frac{\Delta VI\_maximum - \Delta VI}{\Delta VI\_maximum - \Delta VI\_minimum} \tag{3}$$

20



**Table 1.** Photochemical reflectance index (PRI) formulations and structural and water vegetation indices used in this study. R is the reflectance at the specified wavelength in nm.

| Index | Equation | Reference |
|-------|----------|-----------|
| $PRI_{570}$ | (R531-R570)/(R531+R570) | Gamon, Peñuelas, and Field (1992) |
| PRI1 | (R528-R567)/(R528+R567) | Gamon, Filella, and Penuelas (1993) |
| PRI2 | (R539-R570)/(R539+R570) | Penuelas, Filella, and Gamon (1995) |
| PRI3 | (R531-R515)/(R531+R515) | Hernández-Clemente et al. (2011) |
| PRI4 | (R531-R512)/(R531+R512) | Hernández-Clemente et al. (2011) |
| PRI5 | (R531-R600)/(R531+R600) | Gamon, Filella, and Penuelas (1993) |
| PRI6 | (R531-R670)/(R531+R670) | Gamon, Filella, and Penuelas (1993) |
| PRI7 | RDVI=(R800-R670)/(R800+R670) ^0.5 PRI570/[RDVI*(R700/R670)] | Zarco-Tejada et al. (2013) |
| NDVI | (R800-R680)/(R800+R680) | Tucker (1979) |
| WI | R900/R970 | Peñuelas et al. (1993) |
| WI/NDVI | R900/R970/((R800-R680)/(R800+R680)) | Peñuelas and Inoue (1999) |
| mNDVI705 | (R750-R705)/(R750+R705-2*R445) | Sims and Gamon (2002) |

### 2.4. Statistical Analysis

Measurements taken from pots that had the same water treatments were averaged and used in the analysis. Pearson correlation coefficient (r) measured the linear correlation among VIs, shadow fractions, and RWC. Coefficient of variation (CV) and standard deviation were used to evaluate the variation of observations. Least-square linear regression model was used to describe the relationship between VIs and RWC. $R^2$ was used to evaluate the significance of the empirical relationship, and root mean square error (RMSE) was used to measure the actual average differences between measurements and modeled predictions. Statistical analysis was performed in the MATLAB software.

### 3 Results

### 3.1 PRI of stressed and unstressed plants

The spectra of sunlit-leaf pixels and shadow pixels are presented in Figure 2. The reflectance of the shadow was obviously lower than that of the leaf. The spectra of the shadow showed a rise in the near infrared region, but unlike the spectra of the leaf, the peak in the green region was not as obvious as that in the sunlit leaves.





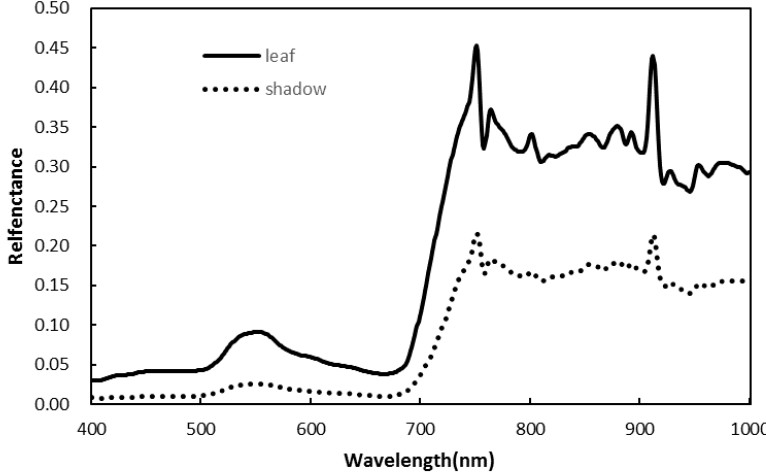

**Figure 2.** Spectra of sunlit leaves and shadow.

We calculated the difference between VI of sunlit leaves and VI of shadow for stressed and unstressed plants separately. For each VI, the statistical summary demonstrated pronounced differences between stressed and unstressed plants (Table 2).

5   For unstressed plants, △PRI570, △PRI1, △PRI2, △NDVI, and △mNDVI705 had higher values in shadow than in sunlit leaves, but the rest of the studied VIs had lower values in shadow than in sunlit leaves. For stressed plants, different formulations of PRI had lower values in shadow than in sunlit leaves, except for △PRI7 and △WI/NDVI.

**Table 2.** The maximum, minimum, mean, coefficient of variation (CV) and range of the difference between vegetation
10   indices (VIs) of sunlit leaves and VIs of shadow (ΔVI) for unstressed plants (**a**) and stressed plants (**b**).

(**a**)

|  | Maximum | Minimum | Mean | CV | Range |
|---|---|---|---|---|---|
| △PRI570 | 0.0136 | -0.0408 | -0.0170 | -1.0005 | 0.0544 |
| △PRI1 | -0.0021 | -0.0395 | -0.0244 | -0.5350 | 0.0374 |
| △PRI2 | 0.0260 | -0.0298 | -0.0047 | -3.4747 | 0.0558 |
| △PRI3 | 0.0418 | -0.0155 | 0.0161 | 1.0059 | 0.0573 |
| △PRI4 | 0.0439 | -0.0223 | 0.0118 | 1.5975 | 0.0662 |
| △PRI5 | 0.0825 | -0.0298 | 0.0197 | 1.7388 | 0.1124 |
| △PRI6 | 0.1960 | -0.0157 | 0.0844 | 0.7125 | 0.2117 |
| △PRI7 | 0.0155 | -0.0129 | 0.0044 | 1.7887 | 0.0283 |
| △NDVI | 0.0534 | -0.0719 | -0.0168 | -2.0570 | 0.1253 |
| △WI | 0.0656 | -0.0453 | 0.0028 | 1.0522 | 0.1109 |
| △WI/NDVI | 0.1194 | -0.0179 | 0.0295 | 1.4640 | 0.1373 |
| △mNDVI705 | 0.0574 | -0.0795 | -0.0282 | -1.4538 | 0.1369 |





(**b**)

|  | Maximum | Minimum | Mean | CV | Range |
|---|---|---|---|---|---|
| ΔPRI570 | 0.0530 | -0.0273 | 0.0154 | 1.2558 | 0.0802 |
| ΔPRI1 | 0.0306 | -0.0364 | 0.0015 | 2.7709 | 0.0670 |
| ΔPRI2 | 0.0602 | -0.0150 | 0.0274 | 0.7380 | 0.0752 |
| ΔPRI3 | 0.0811 | 0.0018 | 0.0387 | 0.4814 | 0.0792 |
| ΔPRI4 | 0.0943 | -0.0093 | 0.0426 | 0.5510 | 0.1036 |
| ΔPRI5 | 0.1506 | 0.0081 | 0.0766 | 0.5168 | 0.1425 |
| ΔPRI6 | 0.3110 | 0.0240 | 0.1853 | 0.4076 | 0.2870 |
| ΔPRI7 | 0.0088 | -0.3965 | -0.0999 | -1.0143 | 0.4053 |
| ΔNDVI | 0.2361 | -0.0386 | 0.0901 | 0.8437 | 0.2747 |
| ΔWI | 0.0829 | -0.0124 | 0.0434 | 0.5770 | 0.0952 |
| ΔWI/NDVI | 0.2395 | -1.2349 | -0.3171 | -1.1939 | 1.4744 |
| ΔmNDVI705 | 0.1273 | -0.0872 | 0.0414 | 1.3388 | 0.2145 |

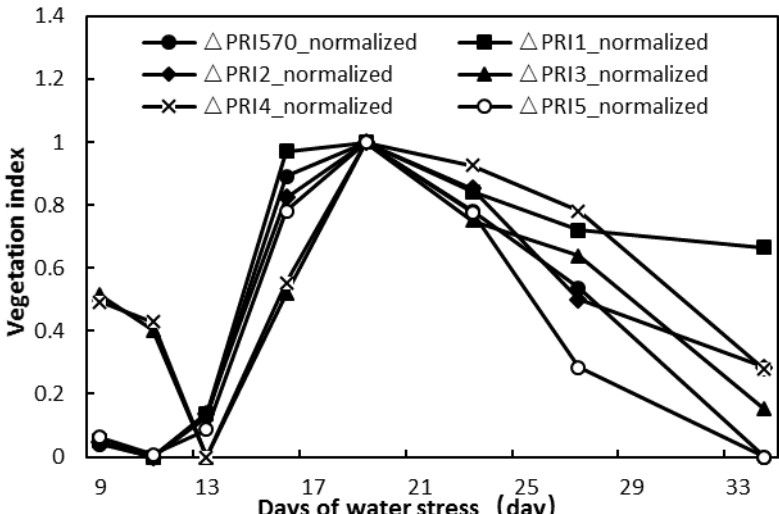

**Figure 3.** Temporal variation of the normalized difference between VI of sunlit leaves and VI of shadow during the water stress treatment.

To analyze variations of ΔVI during water stress, we normalized ΔVI using the maximum and minimum ΔVI. As illustrated in Figure 3, the normalized ΔPRI570, ΔPRI1, ΔPRI2 and ΔPRI5 show similar temporal trends after irrigation stopped. They fell on the ninth to eleventh day of the water-deficit stress treatment, increased drastically till the nineteenth day, and then decreased till the senescence. The performance of ΔPRI3 was roughly consistent with the performance of ΔPRI4. They fell on the ninth to thirteenth days of the water-deficit stress treatment, rose to the maximum on the nineteenth day, and then decreased after that.



Both PRI_ leaf and PRI_shadow were sensitive to RWC. Take PRI570 for example, PRI570 of stressed plants declined as water resource became limiting after irrigation stopped. Interestingly, during the first thirteen days after irrigation stopped, PRI570 of stressed plants slightly increased, indicating the photosynthesis was not impacted by mild water stress. PRI570_shadow was obviously higher than PRI570_mixed and PRI570_leaf during the first thirteen days after irrigation

stopped, but PRI570_shadow was significantly lower than PRI_leaf and PRI_mixed after the thirteenth day. By the twenty-seventh to thirty-fourth days, when leaves dried out, the values of PRI570 reached the similar low values for the leaf pixels, mixed pixels and shadow pixels. Compared with the PRI570 of stressed plants, the time series PRI570 of unstressed plants during the same time as the water stress treatment (Figure 4b) was relatively stable. As expected, PRI570_shadow was higher than PRI570_leaf and PRI570_mixed during the studied period.

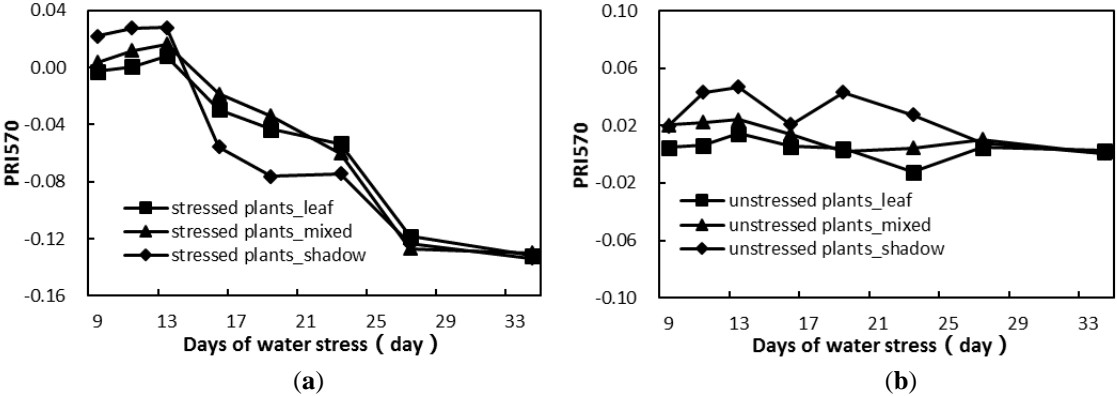

**Figure 4.** Time series of photochemical reflectance index (PRI570) of sunlit leaves, shadow, and the mixture of leaves and shadow for stressed plants (**a**) and unstressed plants(**b**).

### 3.2 The shadow effect on water stress detection

To assess the shadow effect on detecting water stress with PRI, we mixed different fractions of shadow reflectance with sunlit-leaf reflectance, and analyzed the relationship between RWC and VIs calculated from the mixed reflectance of shadow and sunlit leaves. PRI7 and WI/NDVI were negatively correlated with RWC (Table 3), and the rest of VIs were positively correlated with RWC. Among the studied VIs, PRI570, PRI1, PRI2, PRI5 and PRI7 demonstrated better performances in predicting RWC of winter wheat. Although the ratio of sunlit leaves to shadow was different, the correlation coefficient

between RWC and VIs did not show pronounced variations accordingly. Figure 5 shows the standard deviation of the correlation coefficients shown in Table 3. Compared with the other VIs, WI, PRI7, PRI6, PRI4 and PRI3 had higher standard deviations, indicating that shadow had a greater influence on the capability of these VIs in monitoring water stress.

Figure 6 illustrated examples of the significant relationships between water stress indicators and PRI calculated from PRI_leaf and PRI_mixed, respectively. PRI570 and PRI2 did not show obvious differences between sunlit-leaf pixels and

mixed pixels in terms of their relationships with RWC.  PRI7_leaf provided more accurate estimates of RWC than PRI7_mixed did.



We further analyzed the impact of shadow fractions on the slope and intercept of the linear regression equation between VIs and water stress indicators (Table 4). The slope and intercept of the studied VIs were strongly correlated with shadow fractions, except PRI4 (Table 4). Examples of the correlation between shadow fractions and the parameters of the linear regression equation are demonstrated in Figure 7. The slope and intercept of the linear regression equations based on PRI5

5 and NDVI had the most significant correlation with shadow fractions. The slope of PRI4 was not correlated with shadow fractions. We also noticed that PRI1, PRI3, PRI4, WI, and mNDVI705 demonstrated minimal variations in the values of slope and intercept, indicating their insensitivities to varying shadow fractions in the application of water stress detection.

To evaluate if these changes in the values of linear regression parameters affected the accuracy of RWC estimates, we applied the regression function derived from the VIs of the generally applicable sunlit leaves/shadow ratio of 50/50 to detect

10 water stress using VIs of the varying sunlit leaves/shadow ratio. Results showed that RMSE of RWC estimates did not vary significantly with shadow fractions, implying the minimal impact of varying shadow fractions on water stress detection using PRI.



**Table 3.** Relationships between relative water content (RWC) and vegetation indices (VIs) calculated by reflectance of varying ratio of sunlit leaves to shadow in winter wheat during the study period.

| RWC | mixed | Shadow 0% leaf100% | Shadow 10%leaf 90% | Shadow 20%leaf 80% | Shadow 30%leaf 70% | Shadow 40%leaf 60% | Shadow 50%leaf 50% | Shadow 60%leaf 40% | Shadow 70%leaf 30% | Shadow 80%leaf 20% | Shadow 90% leaf10% | Shadow 100% leaf0% |
|---|---|---|---|---|---|---|---|---|---|---|---|---|
| PRI570 | 0.65** | 0.65** | 0.65** | 0.65** | 0.65** | 0.65** | 0.65** | 0.65** | 0.64** | 0.64** | 0.64** | 0.63** |
| PRI1 | 0.61** | 0.58** | 0.58** | 0.59** | 0.59** | 0.60** | 0.60** | 0.61** | 0.62** | 0.62** | 0.63** | 0.64** |
| PRI2 | 0.65** | 0.64** | 0.64** | 0.64** | 0.64** | 0.64** | 0.64** | 0.64** | 0.64** | 0.63** | 0.63** | 0.63** |
| PRI3 | 0.52** | 0.44** | 0.45** | 0.45** | 0.46** | 0.47** | 0.47** | 0.48** | 0.49** | 0.49** | 0.49** | 0.50** |
| PRI4 | 0.42** | 0.38** | 0.39** | 0.40** | 0.40** | 0.41** | 0.42** | 0.43** | 0.43** | 0.44** | 0.45** | 0.46** |
| PRI5 | 0.62** | 0.61** | 0.61** | 0.61** | 0.61** | 0.61** | 0.62** | 0.62** | 0.62** | 0.62** | 0.62** | 0.61** |
| PRI6 | 0.48** | 0.40** | 0.46** | 0.47** | 0.48** | 0.48** | 0.49** | 0.50** | 0.50** | 0.51** | 0.51** | 0.52** |
| PRI7 | -0.62** | -0.67** | -0.59** | -0.59** | -0.59** | -0.59** | -0.59** | -0.59** | -0.59** | -0.54** | -0.58** | -0.57** |
| NDVI | 0.53** | 0.50** | 0.50** | 0.51** | 0.51** | 0.51** | 0.52** | 0.52** | 0.52** | 0.52** | 0.53** | 0.53** |
| WI | 0.40** | 0.30* | 0.31* | 0.32** | 0.34** | 0.35** | 0.36** | 0.38** | 0.39** | 0.41** | 0.42** | 0.44** |
| WI/NDVI | -0.53** | -0.51** | -0.50** | -0.51** | -0.51** | -0.51** | -0.52** | -0.52** | -0.52** | -0.52** | -0.52** | -0.51** |
| mNDVI705 | 0.51** | 0.51** | 0.51** | 0.51** | 0.51** | 0.52** | 0.52** | 0.52** | 0.52** | 0.53** | 0.53** | 0.53** |

**. Correlation coefficient significant at P＜0.01.

5    *. Correlation coefficient significant at P＜0.05.





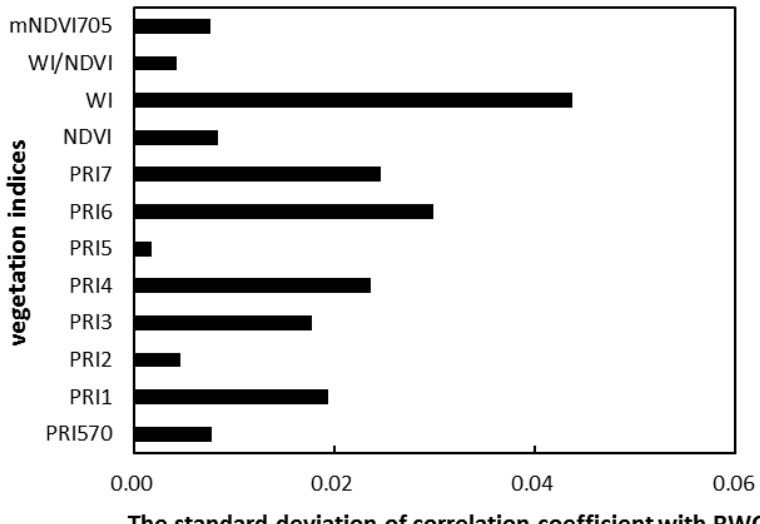

**Figure 5.** The rang of the change in correlation coefficient between relative water content (RWC) and vegetation indices (VIs).

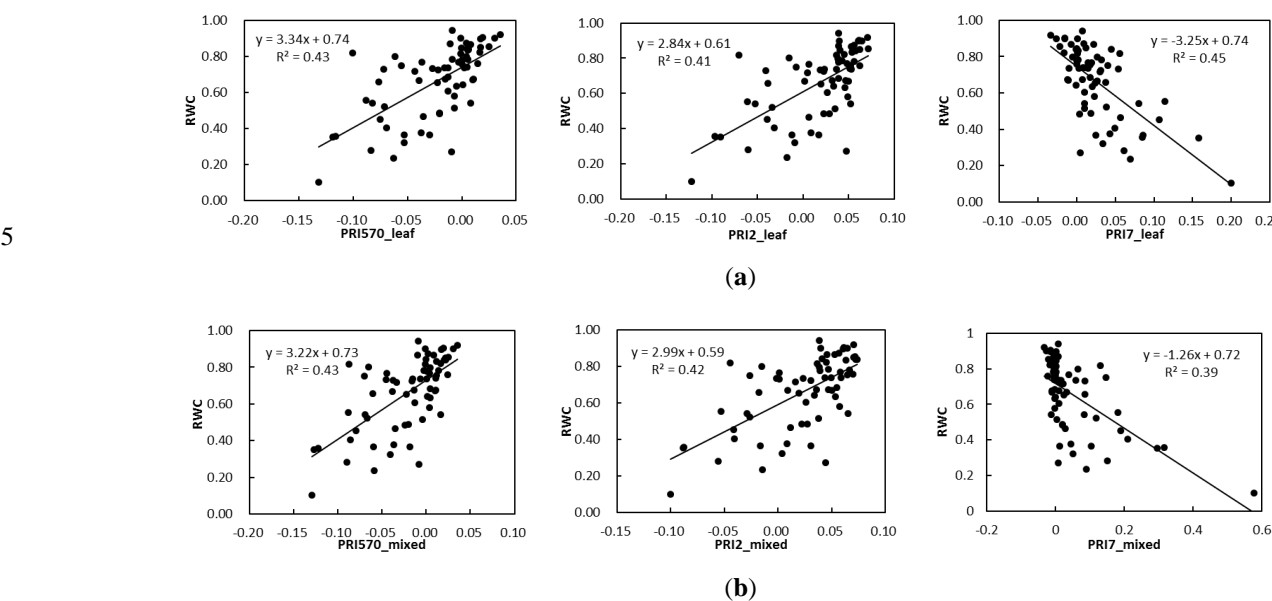

**Figure 6.** Examples of the strong relationship between photochemical reflectance index of sunlit leaves (PRI_leaf) and relative water content (RWC) (**a**) and the relationship between photochemical reflectance index of mixture of sunlit leaves and shadow (PRI_mixed) and RWC (**b**).





**Table 4.** Relationships between shadow fractions and the slope and intercept of the linear regression equation between VIs and water stress indicators.

| VI | Slope | Intercept |
|---|---|---|
| PRI570 | 0.98** | -0.93** |
| PRI1 | 0.94** | -0.77** |
| PRI2 | 0.98** | 0.99** |
| PRI3 | 0.64* | 0.90** |
| PRI4 | 0.00 | 0.80** |
| PRI5 | 0.98** | 0.99** |
| PRI6 | 0.94** | 0.98** |
| PRI7 | 0.99** | 0.99** |
| NDVI | -0.98** | 0.98** |
| WI | 0.97** | -0.96** |
| PWI | 0.93** | -0.90** |
| mNDVI705 | -0.94** | 0.95** |

**. Correlation coefficient significant at P＜0.01.

*. Correlation coefficient significant at P＜0.05.

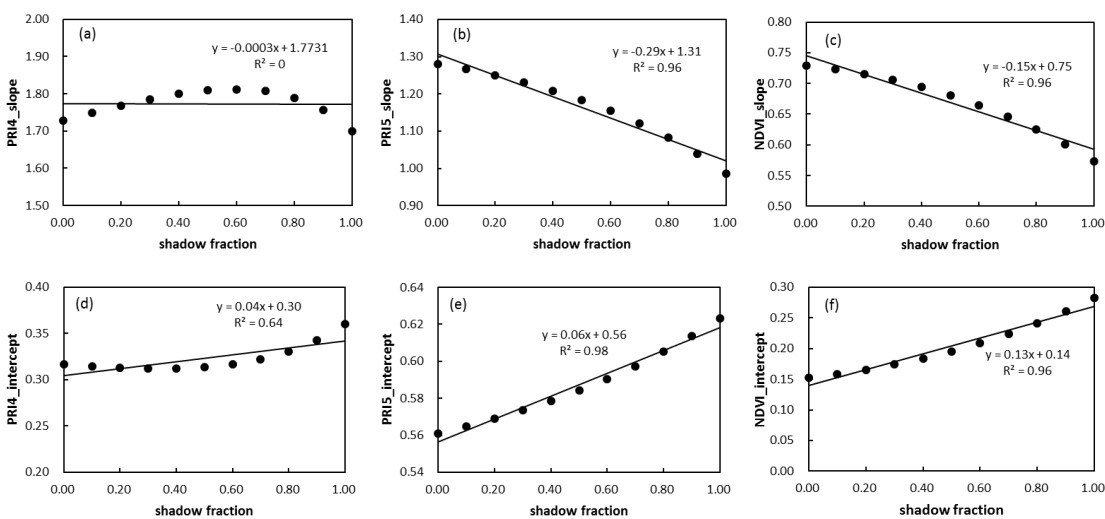

**Figure 7.** Relationships between shadow fractions and the slope of the regression equation of relative water content (RWC) and photochemical reflectance index (PRI) PRI3 (**a**), PRI5 (**b**), and normalized difference vegetation index (NDVI) (**c**); and

10   relationships between shadow fractions and the intercept of the regression equation of RWC and PRI3 (**d**), PRI5 (**e**), and NDVI (**f**).



## 4. Discussion References

Theoretically, sunlit leaves are more likely to experience high light-induced environmental stress than shaded leaves (Hilker et al., 2008; Middleton et al., 2009; Cheng et al., 2012). Data from previous field samplings and the model simulations, although limited, confirmed the impact of shadow fractions on PRI values (Middleton et al., 2009; Cheng et al.,
2012; Takala and Mõttus, 2016). While interests of detecting water stress of plants with PRI are increasing, studies rarely analyzed the shadow effects on the performance of PRI in water stress detection. This study evaluated the impact of the varying shadow fractions on water stress detection in winter wheat using PRI derived from hyperspectral images.

Our results showed that for unstressed plants PRI570, PRI1, and PRI2 of shadow was generally higher than those of sunlit leaves, indicating that the intensity of xanthophyll-regulated photo-protection is lower than in sunlit leaves (Hilker et
al., 2008; Middleton et al., 2009; Cheng et al., 2012). The difference between PRI_shadow and PRI_leaf varied with PRI formulations. ΔPRI570 ranged from -0.048 to 0.0136, roughly agreed with results presented in previous studies. Takala and Mõttus (2016) reported the range of ΔPRI without a shadow correction was -0.01 – 0.10 at the boreal forest. Middleton et al. (2009) reported ΔPRI of -0.035 at a Douglas-fir forest in Canada. Cheng et al. (2012) demonstrated that the average PRI values varied from -0.008 to 0.005 for sunlit leaves and from 0.002 to 0.022 for shaded leaves measured in the corn field.
Mõttus et al. (2015) presented the difference between canopy PRI (including PRI of shaded leaves) and PRI of sunlit leaves ranged from -0.025 to 0.073 for pine, spruce and birch.

Interestingly, for stressed plants, PRI570, PRI1, and PRI2 of shadow was lower than those of sunlit leaves, which was contrary to the findings for unstressed plants. Also, the range of the difference between PRI_leaf and PRI_shadow increased as water stress progressed, but it then decreased to the minimum when prolonged drought caused premature senescence. We
speculated that one reason might be a sustained water stress deficit inducing chlorophyll degradation more severely on old leaves (Bolhar-Nordenkampf, Hofer, and Lechner, 1991; Ciganda, Gitelson, and Schepers, 2012; Liu et al., 2015), which are shaded by new leaves above. As several studies showed that PRI was related to the pigment content (Suárez et al., 2009; Gitelson, Gamon, and Solovchenko, 2017), the early degradation of chlorophyll content in the shaded leaves may lead to the change of PRI values. However, both the new leaves and old leaves eventually wilt after the prolonged water stress, resulting
in the decreasing range in the difference between PRI_leaf and PRI_shadow after 19 days of water stress treatment.

Although the values of PRI_shadow were different from PRI_leaf for both unstressed and stressed plants, the effect of the varying fractions of shadow did not lead to the substantial change in the significance of the relationship between PRI and water stress indicators. We hypothesized that as water resource became more limiting during the treatment the changes in leaf area and pigment content may have a much greater impact on PRI than the de-epoxidation of xanthophyll cycle did. We
tested different formulations of PRI that were proved to minimize the effect of the structural change in canopies in previous studies(Hernández-Clemente et al., 2011; Zarco-Tejada et al., 2013), but they did not show any competitive advantage over PRI570. Another reason that may attribute to the minimal effect of varying shadowing fractions on water stress detection using PRI was the varying degree of shadow. In this study, the reflectance of shadow was derived from pixels classified as



the shadow in the hyperspectral images, instead of searching for the darkest spots in the image, which were possibly the soil background. The less shaded leaves that were classified as the shadow may be more similar to the sunlit leaves than the deeply shaded leaves, resulting in the strong correlation of PRI_shadow with RWC.

Since shadow fractions influenced PRI values, the parameters of the linear relationship between PRI and RWC were
correlated to the shadow fraction. However, we noticed for most studied VIs, the values of intercept and slope of regression models changed within a small range. Furthermore, the varying shadow fraction did not have significant impacts on the accuracy of RWC estimated with PRI. Among the studied VIs, PRI570 provided the most accurate estimates of RWC regardless of shadow fractions. In comparison with the structural VIs and water indices, most formulations of PRI showed better performances in detecting water status in winter wheat, which agreed with results of several previous studies. Suárez
et al.(2010), Rossini et al.(2013), Zarco-Tejada et al. (2013) demonstrated that PRI was more sensitive to changes in water status of different species than NDVI was. Katsoulas et al. (2016) supported that NDVI at 800 and 680 nm was not very sensitive to environmental conditions variations. Schlemmer (2005) reported that the reflectance of stressed plants was increased in the near infrared region due to radiation scattered by air content risen in sponge cavities. Peñuelas et al. (1993) observed a significant decrease in the magnitude of the whole near infrared reflectance of stressed plants only when the plant
was close to wilting. Although the imagery spectrometer used in this study did not cover mid-infrared region which is related with water and lignin content in vegetation (Asner 1998), previous studies indicated that the use of mid-infrared region is insufficient to estimate the leaf water status due to the fact that reflectance changes within a biologically meaningful range are too insignificant and the light signal at that spectrum has high light signal noise (Cordon and Lagorio, 2007; Sun et al., 2008).

**5. Conclusion**

This study evaluated the impact of the varying shadow fraction on seasonal water stress detection in winter wheat using different formulations of PRI. Results demonstrated that PRI570, PRI1, and PRI2 of shadow were higher than those of sunlit leaves for unstressed plants, but the contrary results were achieved for stressed plants. The range of the difference between PRI_leaf and PRI_shadow increased as water stress progressed, but it then decreased to the minimum when prolonged
drought caused premature senescence. Despite the difference between PRI_shadow and PRI_leaf, the significance of the linear relationship between RWC and different formulations of PRI did not show obvious variations with shadow fractions. For most studied PRI formulations, the slope and intercept of the linear regression equation between RWC and PRI changed proportionally with the shadow fraction. However, applying a uniform model to the VIs calculated with varying shadow fractions did not significantly affect the accuracy of RWC predictions. Thus, we concluded that the effect of varying shadow
fractions was minimal to the seasonal water stress detection in winter wheat.



*Author Contributions*. All authors participated to the measurement collection that was part of an intensive field campaign. Xin Yang wrote the manuscript supported in both data interpretation and manuscript writing by Shishi Liu. Yinuo Liu was responsible for the experimental field management and planned the experimental design with Xifeng Ren. A significant contribution to the field data interpretation was also provided by Hang Su.

*Competing interests.* The authors declare no conflict of interest.

*Acknowledgments.* This research was supported by grants from the National Natural Science Foundation of China (Grant No. 41501367) and National Key Research and Development Program of China (Grant No. 2017YFD0100802).

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
