# Peer review of "Assessing shaded-leaf effects on photochemical reflectance index (PRI) for water stress detection in winter wheat"

_Biogeosciences, 2018_

## Referee Comment (RC1) · Anonymous Referee #1 · 22 Nov 2018

Yang et al. studied the impact of shadow fraction on the capability of PRI to predict relative water content, RWC, in winter wheat. The concept of this analysis is very interesting. However, I have some questions on how the authors concluded that a varying shadow fraction has limited impact on the prediction of RWC.

Specific Comments

The conclusion is that shadow fraction does not significantly affect the prediction capabilities of PRI of relative water content. I find that hard to believe. If PRI is different for a range of shadow fractions and water content in an entire plant is generally similar than there must be a difference? Let's think about a pixel with either 10% shadow or 90%

shadow (we don't know what it is), the PRI is different (right?) but the relative water content in the plant is not (or is that a wrong assumption?).

It would be interesting to mention whether this conclusion holds up for other crop types. How generally applicable is the outcome of this research?

I don't understand why the authors focus on just PRI. The title only mentions PRI, but the research also includes different forms of NDVI and WI. However, on Page 3, line 7, only different formulations of PRI are mentioned. It seems like a waste of data, when the feature extraction is so limited. There are many other VI's in literature than can be explored, such as; health index (HI), plant senescing reflectance index (PSRI), renormalized difference vegetation index (RDVI), and normalized photochemical reflectance index (PRIn) to name a couple. A nice overview of narrowband indices can be found in (López-López, Calderón, & González-Dugo, 2016).

The terms "stressed plants" and "mixed" are confusing. I assume this is water stressed, but how stressed are they? It would be cleaner to use quantifying terms like RWC. This also applies to figures 3 and 4. It would be interesting to see the trend over RWC and not over days of water stress. Mixed is mentioned several times and shown in figures, but what is it? What percentage of shadow/sunlit?

A quick google search on "assessing shadow effects on photochemical reflectance index" pointed me at two valuable papers that weren't cited. The authors should consider including them ; (Suárez, Zarco-tejada, Sepulcre-cantó, & Pérez-priego, 2008) and (Zhou et al., n.d.).

Page 1, line 1, the authors of this paper look at more than only PRI. The title makes it seem like this is the only Vegetation Index.

Page 3, line 7, what is the seasonal scale? I don't see this in the rest of the paper.

Page 3, line 30-31, can the authors explain why they use three pots for hyperspectral imaging and three for collecting samples? Why aren't the same pots used for both

imaging and sampling? How big is the possible error that is introduced here?

Page 4, line 1, it is unclear to me what kind of measurements the authors are talking about.

Page 4, line 16-19, can the authors elaborate a little more on the characteristics of the hyperspectral camera. Mention the FWHM (Full Width Half Max) and the spatial resolution.

Page 4, line 21, I don't think you should say "errors", maybe use "variation".

Page 4, line 23, can the authors explain more about the wavelength correction? How is this different from the radiometric calibration and what is the in- and output?

Page 4, line 27, why do the authors use the Mahalanobis distance method, and what is it?

Page 4, line 29, how is the overall accuracy determined? How many validation points are used?

Page 5, line 12-15, the feature extraction in this section is somewhat unclear to me. Can the authors describe why an interpolation function was used and not the closest wavelength to determine the VI's? Moreover, the decision for these VI's is not mentioned and there are other VI's besides PRI. Hence, the title of the paper does not match the content of the paper. There are many other VI's, band combinations and features that can be extracted from a hypercube.

Page 5, lines 15-19, I'm unfamiliar with this approach of normalizing the difference of a normalized index. It would be helpful if the authors show previous studies that use this approach, or if they could justify this method in another way.

Page 6, lines 4-9, the statistical analysis section is very concise and lacks insight in the process. Can the authors make it clear which method is used for which part of the analysis? Further on in the paper the uniform model is introduced, but this is not

mentioned in the statistical analysis section. Please include this.

Page 7, figure 2, legend; "sunlit leave". The figure would be more informative if the confidence interval was included (I assume that the data shown is the average response?). How much do the two groups differ?

Page 7, Table 2a&b, this table contains a lot of information, but it is unclear what the most important numbers are. Maybe move this to supplements. Another statistic which can be insightful is a 2-sample t-test and find the VI with the lowest p-value for the most significant difference between sunlit leaves and shadow.

Page 9, line 12; how is it possible to have a plot of unstressed plants over days of water stress?

Page 11, table 3, what do the numbers mean? Is it the Pearson's correlation coefficient? Make this clear in the caption.

Page 12, figure 5, this figure is not very informative. Either explain why this figure is useful or take it out.

Page 13, table 4, same comment as for table 3, what is the relationship? Is it a correlation coefficient?

Technical Corrections

Page 1, line 21, I think it should be "and frequent".

Page 1, line 26, it states that monitoring the change in canopy temperature is a direct way to evaluate water stress. Shouldn't this be indirect?

Page 5, line 10, I would leave out the following sentence; "The fraction of shadow and the fraction of sunlit leaves should be summed to 1".

Page 6, line 13, leave out "obviously".

Page 6, line 13, leave out "The spectra of the shadow showed a rise in the near infrared
region, but".

Page 8, line 8&10; use "decreased" instead of "fell".

Page 14, line1, I think it should be "Discussion" and not "Discussion References".

References

López-López, M., Calderón, R., & González-Dugo, V. (2016). Early detection and quantification of almond red leaf blotch using high-resolution hyperspectral and thermal imagery. Remote Sensing. Retrieved from http://www.mdpi.com/2072-4292/8/4/276/htm

Suárez, L., Zarco-tejada, P. J., Sepulcre-cantó, G., & Pérez-priego, O. (2008). Assessing canopy PRI for water stress detection with diurnal airborne imagery, 112, 560–575. https://doi.org/10.1016/j.rse.2007.05.009

Zhou, K., Deng, X., Yao, X., Tian, Y., Cao, W., & Zhu, Y. (n.d.). Assessing the Spectral Properties of Sunlit and Shaded Components in Rice Canopies with, 1–17. https://doi.org/10.3390/s17030578

---

## Author Comment (AC1) · 27 Nov 2018

1. "The conclusion is that shadow fraction does not significantly affect the prediction capabilities of PRI of relative water content. I find that hard to believe. If PRI is different for a range of shadow fractions and water content in an entire plant is generally similar then there must be a difference? Let's think about a pixel with either 10% shadow or 90% (we don't know what it is), the PRI is different (right?) but the relative content in the plant is not (or is that a wrong assumption?)." We understand the review's concern. This study aims to evaluate the effect of the shadow fraction on the remote estimation of crop water status using PRI, because shadow is mixed with sunlit leaves in a pixel of

airborne or spaceborne remote sensing images. However, when we collect the ground 'truth', the best way to evaluate water status of crops is to measure the relative water content (RWC) and/or water potential and/or canopy temperature of the sampled plants and then average the values of sampled plants. This value represents the real water status of crops no matter how much fraction of the shadow exists in a pixel. Vegetation indices derived from remotely sensed images are expected to be correlated with the ground measurements, so that we can evaluate the water status in pixels without ground measurements. However, shadow exists in every vegetation pixel and the fraction may vary with pixels and also with image resolution as well as viewing geometry. Thus, our analysis was trying to figure out whether the varying shadow fractions may influence PRI and its ability to estimate RWC. If the shadow effect was influential, and then the mixture analysis is needed before applying PRI to evaluate water status. But if the shadow effect was minimal (as the conclusion of our analysis), and we can ignore the varying shadow fractions among pixels. We changed the shadow fraction from 10% to 90% in order to demonstrate the relationship between shadow fractions and PRI, but in reality, the range of shadow fraction may not be so large. We thought the assumption of this study is correct, in agreement with the sampling strategy in the studies of remote detection of plant water stress, but we welcome comments, suggestions, and arguments.

2. "It would be interesting to mention whether this conclusion holds up for other crop types. How generally applicable is the outcome of this research?" Thanks for the suggestion. We wouldn't reach the same conclusion for the other crops, because from our reasonable guess it is probably related with canopy structural features, such as leaf orientation and green leaf coverage. Therefore, it is hard to reach a conclusion or propose a hypothesis for the crops with different leaf orientation or green leaf coverage. But it is definitely a good suggestion for our next experiment.

3. "I don't understand why the authors focus just PRI. The title only mentions PRI, but the research also includes different forms of NDVI and WI. However, on page3, line 7,

only different formulations of PRI are mentioned. It seems like a waste of data, when the feature extraction is so limited. There are many other VI's in literature than can be explored..." Thanks for the valuable comments. The audience may have the same concern with you. As mentioned in the introduction section, several methods can be used to remotely assess water status of crops, and relating PRI with physiological water stress indicators of crops is one of them. Unlike the other vegetation indices (VIs), PRI is sensitive to physiological properties of plants, particularly highly sensitive to the changes in photosynthetic rate. Therefore, several studies have been done to evaluate water stress of plants specially using PRI. But in this study, we also included the other VIs that are sensitive to changes in the plant canopy structure, pigment content, and water content for comparisons, in order to show the advantage of PRI in estimating RWC. In this case, we didn't analyze the shadow effect on the other indices. However, we understand the comparisons with the other selected vegetation indices may cause confusions, and we will make revisions accordingly if a major revision is suggested by the editor.

---

## Referee Comment (RC2) · Anonymous Referee #2 · 4 Feb 2019

General Comments: This is a paper that unfortunately suffers from too many flaws to be publishable, in my opinion. The reasons are thus: 1. This study collected PRI data within a window between 1000 and 1400h. Unfortunately, wheat PRI can change dramatically between 1000 and 1400h. Magney et al. (2016) demonstrated that PRI can vary by a factor of 4 between 1000 and 1400h, particularly later in the growing season when water stress is at its peak. This is likely problematic for the current study. It would have been helpful for the authors to conduct an experiment of how the PRI in their wheat plots changes over the course of the data collection period. This information is fundamental to determining whether it is valid to group data across 1000-1400h, or whether the data must be binned in a more time-specific manner before analyzed.

[Figure]

2. Because foliar deepoxidation state will relate to the instantaneous level of non-photosynthetic quenching (NPQ), and NPQ relates to the instantaneous amount of PAR striking a leaf, what is being defined as a sunlit leaf? In figure 1, some leaves are normal to the camera lens, then curve away. There is more of a continuum of light values, rather than two distinct classes of sunlit canopy vs. shaded canopy. As a result, the analysis is flawed because it is trying to capture a process that responds to a continuum (of incident PAR, specifically) using a binary shadow/non-shadow classification. The biological process in question is nonlinear, and the method is oversimplified. 3. The variants of PRI selected for this study are influenced by both long term (constitutive) and short term (facultative) plant physiological processes, and the influences of long term vs. short term pigment pools cannot be isolated from each other. See Gamon and Berry (2012) for more detail. 4. The light (PAR) incident on the plants was not measured or considered in the analyses. This, combined with the fact that the authors are treating a continuous variable of light intensity as a categorical variable (i.e. sunlit vs. shaded) unfortunately are fundamental omissions that make the results of this study invalid in my opinion. Specific Comments: Title: Please remove "the" in the title to make the title more readable. Abstract Line 7: PRI can correlate with several types of plant ecophysiological functions depending on the timeframe of analysis. As a result, several different types of plant stress (nutrient, water, pathogen) can influence the de-epoxidation state of a plant as it responds to excess light. It would also be helpful to indicate here that the PRI is a remotely sensed spectral vegetation index. I therefore suggest that the authors remove 'water' from this first line and rather state "…a pre-visual remotely sensed indicator of plant stress." Line 8: "…whether variations the shadow fraction, which can be influenced by varying view angle and crop density…" Line 11: Three different variants of PRI are presented here, without any indication that the different formulations of PRI can be interpreted in different ways, based on several considerations (e.g. timeframe of analysis, phenological responses within the growth cycle, etc.). Specifying how many formulations were tested would be helpful. For example, "…using 6 different formulations…" (line 10) and then "Results demonstrated

that three of the PRI formulations (PRI570, PRI1, and PRI2). . ." Abstract in general: in a paper looking at the PRI to assess plant stress under different light levels (i.e. shadow fractions) I think it imperative that the authors mention the core mechanism driving changes in PRI at short timescales, which is related to the relationship between excess light and non-photochemical quenching. NPQ is the process by which the plant shunts excess light from a cell's light harvesting complex in the form of heat. Introduction Page 1, Line 23: Suggest new paragraph starting with "Remote sensing. . ." Line 24: ". . .assess water status. . ." Page 2, line 4: Please see first comment I made on the abstract. As I read further in this section, it appears that the authors do a pretty good job of overviewing the connection between PRI and stress. It would be useful to still include some mention of non-water-related stressors here in this section (e.g. nutrient availability, pathogens) and also structure the argument to explain to the reader that the xanthophyll cycle serves to protect the light harvesting complex from excess light, and that the threshold for a plant to deal with excess light varies according to a multitude of environmental factors (water availability being one, yes, but not the only one). Line 25: I respectfully disagree that the impacts of shadowing on PRI are generally ignored. This is a widely recognized phenomenon. The authors should re-word this statement. Line 30: The more mechanistically appropriate way to explain this is that the xanthophyll cycle status is affected by incident PAR, which is in turn affected by the level of self-shading (i.e. shadow fraction) within a canopy. Intro in general: Magney et al. (Remote Sensing of Environment, 2016, 173: 84-97) reported relationships between derivations of the PRI and various environmental conditions (including VPD) in three different portions of a wheat field. This paper should be briefly overviewed in the introduction and potentially the discussion due to the goals of the study being closely related to the current paper. Page 3, Line 2: ". . .the PRI's capability. . ." The PRI does not have agency or capability. Scientists have a capability to interpret PRI data to detect stress. Please reword. Materials and Methods Page 3, Line 13: "rectangular" Page 3, Line 16: "plots" should be "pots" Line 20: remove "a" Line 22: Were all pots located in the rainout shelter? Because plant light harvesting complexes and pigments

can change as a result of the ambient light environment in which they are growing, it is important to know whether the rain-out shelters reduced the PAR striking the canopy. What was the influence of these shelters on the ambient light condition? Line 23 and 26: "control" is more specific than "reference" Page 4, Line 20: Due to a number of factors such as VPD and air temperature, the PRI of wheat can change dramatically between 1000 and 1400h. See Magney et al. 2016, who demonstrated that PRI can vary by a factor of 4 between 1000 and 1400, particularly later in the growing season. This could be very problematic for the current study. Did the authors conduct an experiment of how the PRI in their wheat plots changes over the course of the data collection period? This information is fundamental to determining whether it is valid to group data across 1000-1400h, or whether the data must be binned in a more time-specific manner before analyzed. Line 30: Depending on the specs of this journal, you probably need to specify the ENVI manufacturer, version, etc. Figure 1: This classified image highlights a question for me. Because the deepoxidation state will relate to the instantaneous level of NPQ, and NPQ relates to the instantaneous amount of PAR striking a leaf, what is being defined as a sunlit leaf? In figure 1, some leaves are normal to the camera lens, then curve away. There is more of a continuum of light values, than two distinct classes of sunlit canopy vs. shaded canopy. As a result, the analysis may be flawed because it is trying to capture a process that responds to a continuum (of incident PAR, specifically) using a binary shadow/non-shadow classification. Table 1: Unfortunately, of all of the PRI calculations used in this study, the two variants of PRI that have been shown to correlate most strongly with water status and other diurnally changing physiological variables, the deltaPRI and the PRIo, (Magney et al. 2016) were not calculated or used in this study. The various variants of PRI used in this particular study are influenced strongly by longer-term chlorophyll:carotenoid ratios that will mask the instantaneous effects of changing light or VPD conditions (Gamon and Berry, 2012). (Note: the deltaPRI calculations shown in Table 2 are different from deltaPRI in the literature and will not remove the seasonal effect of pigment phenology).

Ref: Gamon, J.A., & Berry, J.A. (2012). Facultative and constitutive pigment effects

on the photochemical reflectance index (PRI) in sun and shade conifer needles. Israel Journal of Plant Sciences, 60(1), 85–95.

Final comment: in my reading of this paper, the light (PAR) incident on the plants was not measured or considered in the analyses. This, combined with the fact that the authors are treating a continuous variable of light intensity as a categorical variable (i.e. sunlit vs. shaded) unfortunately are fundamental omissions that make the results of this study invalid in my opinion.
* * *

---

## Author Comment (AC2) · 7 Feb 2019

1. This study collected PRI data within a window between 1000 and 1400h. Unfortunately, wheat PRI can change dramatically between 1000 and 1400h. Magney et al. (2016) demonstrated that PRI can vary by a factor of 4 between 1000 and 1400h, particularly later in the growing season when water stress is at its peak. This is likely problematic for the current study. It would have been helpful for the authors to conduct an experiment of how the PRI in their wheat plots changes over the course of the data collection period. This information is fundamental to determining whether it is valid to group data across 1000-1400h, or whether the data must be binned in a more

time-specific manner before analyzed.

Thanks for the suggestion. PRI changes diurnally driven by the xanthophyll cycle and also changes seasonally driven by the variations in pigment pool sizes (e.g. carotenoid/chlorophyll). Since this study focused on evaluating seasonal water stress, we conducted the experiment during the noon. Although we did not conduct the diurnal measurements, we recorded the time when each hyperspectral image was taken. Thus, if the major revision was allowed, we can filter the data to make sure all the data used for analyses were collected within an hour.

2. There is more of a continuum of light values, rather than two distinct classes of sunlit canopy vs. shaded canopy. As a result, the analysis is flawed because it is trying to capture a process that responds to a continuum (of incident PAR, specifically) using a binary shadow/non-shadow classification. The biological process in question is nonlinear, and the method is oversimplified.

We agreed with the reviewer's opinion that PRI is related to the irradiance, but it is hard to measure the irradiance received by each leaf, and thus to analyze the difference between the sunlit and shaded leaves provides a feasible way to evaluate the impact of the vertical illumination distribution and mixed pixels issue. From a remote sensing perspective, shadow or shade is inevitable in a pixel, and we usually assume the contribution of vegetation and shade to the reflectance of the pixel is linear (Dennison and Roberts 2003; Tane et al. 2018). The shade fraction may vary from pixel to pixel depending on the viewing zenith angle, solar angle, and the vegetation fraction. Therefore, this study aimed to evaluate the impact of shade fractions on the assessment of seasonal water stress using PRI. Although the incident PAR is continuously distributed within the canopy, sunlit leaves and shade have contrasting illumination in a pixel. Several studies have analyzed the differences of PRI in sunlit leaves and shaded leaves, and the sunlit leaves are usually selected/defined as the top of the canopy leaves that receive high irradiance, and the shaded leaves are selected/defined as the leaves at the bottom of the canopy that receive low irradiance(Gamon and Berry 2012; Takala

and Mõttus 2016). Takala and Mõttus
(2016) picked the darkest and brightest top-of-crown pixel in the aerial photo, based on the broadband reflectance factor value as the samples of sunlit and shaded leaves. Instead of manually picked the sunlit and shaded leaves, we classified the hyperspectral images into sunlit and shaded leaves, which have distinct spectra. If manually selecting the darkest and brightest leaves was more appropriate than classification, we would like to make a change accordingly.

3. The variants of PRI selected for this study are influenced by both long term (constitutive) and short term (facultative) plant physiological processes, and the influences of long term vs. short term pigment pools cannot be isolated from each other. See Gamon and Berry (2012) for more detail.

We agreed with the reviewer that PRI is influenced by both constitutive and facultative plant physiological processes, and these two processes are hard to be isolated from each other. Although we did not measure the xanthophyll cycle, we did measure the chlorophyll and carotenoid content. And our analyses showed that PRI was strongly related to the ratio of chlorophyll to carotenoid (not shown in the study). If the major revision was allowed, we would add the analysis of the correlation between PRI and pigment pool sizes. We would also like to try to use the difference between the PRI of sunlit leaves (brightest pixels) and shaded leaves (darkest leaves) to minimize the constitutive effects. Although the difference between the PRI of sunlit and shaded leaves is different from the delta PRI proposed by Gamon and Berry (2012) and Magney et al. (2016), the PRI of shaded leaves may be a proxy for the PRI at epoxidation state, since shaded leaves do not experience de-epoxidation of xanthophyll cycle as the sunlit leaves do. Also Hwang found that the ratio of PRI in sunlit canopy (backward direction MODIS images) to PRI in shaded canopy (forward direction images) provided better correlations with drought signal. The test of the difference between the PRI of sunlit and shaded leaves in the water stress assessment may provide insights into the applications of multi-angle aerial or satellite images in monitoring crop water stress.

4. The light (PAR) incident on the plants was not measured or considered in the analyses.

Thanks for the reviewer's comment. We did not realize the importance of PAR in the analyses of PRI, as our focus was on the difference between PRI in sunlit and shaded leaves and their correlations with RWC. The weather station near the study site takes PAR measurements. And also the DN value in near infrared bands of the gray panel can be used as the proxy of PAR. We will evaluate the relationship between PRI and PAR for pots under different levels of water stress, if the major revision was allowed.

References:

Dennison, Philip E., and Dar A. Roberts. 2003. "Endmember Selection for Multiple Endmember Spectral Mixture Analysis Using Endmember Average RMSE." Remote Sensing of Environment 87 (2–3): 123–35. https://doi.org/10.1016/S0034-4257(03)00135-4.

Gamon, John A., and Joseph A. Berry. 2012. "Facultative and Constitutive Pigment Effects on the Photochemical Reflectance Index (PRI) in Sun and Shade Conifer Needles." Israel Journal of Plant Sciences 60 (1): 85–95. https://doi.org/10.1560/IJPS.60.1-2.85.

Takala, Tuure L.H., and Matti Mõttus. 2016. "Spatial Variation of Canopy PRI with Shadow Fraction Caused by Leaf-Level Irradiation Conditions." Remote Sensing of Environment 182 (September): 99–112. https://doi.org/10.1016/j.rse.2016.04.028.

Tane, Zachary, Dar Roberts, Sander Veraverbeke, Ángeles Casas, Carlos Ramirez, and Susan Ustin. 2018. "Evaluating Endmember and Band Selection Techniques for Multiple Endmember Spectral Mixture Analysis Using Post-Fire Imaging Spectroscopy." Remote Sensing 10 (3): 389. https://doi.org/10.3390/rs10030389.

---

## Author Response (AR1)

Response to Reviewer #1

Dear reviewer,

We greatly appreciate the reviewers' comments and suggestions, which significantly improve our study. We did a major revision as suggested. Basically, we selected images captured between 12:00-14:00pm as suggested, and thus rerun all the analysis. Minor revisions can be tracked in word document, and major revisions was conducted in several sections highlighted with yellow. Below are the point-by-point responses:

1. **The conclusion is that shadow fraction does not significantly affect the prediction capabilities of PRI of relative water content. I find that hard to believe. If PRI is different for a range of shadow fractions and water content in an entire plant is generally similar than there must be a difference? Let's think about a pixel with either 10% shadow or 90% shadow (we don't know what it is), the PRI is different (right?) but the relative water content in the plant is not (or is that a wrong assumption?).**

   The fraction of shaded leaves may vary with pixels due to the viewing geometry, species, plant densities, etc., but studies usually ignored the impacts of shaded leaves and adopted the uniform sampling strategy. For example, in Panigada et al. (2014) and Rossini et al.'s (2013) study, measurements of physiological properties of plants were conducted on the youngest fully expanded leaf of crops; in Rapaport et al.' (2015) study, measurements were conducted on a single sun-exposed, youngest fully-matured leaf of each grapevine. Therefore, we changed the fractions of shaded leaves and evaluated its impact on water stress detection using the uniform RWC sampling strategy, as conducted in most studies.

2. **It would be interesting to mention whether this conclusion holds up for other crop types. How generally applicable is the outcome of this research?**

   As canopy geometry and leaf orientation may have great impacts on the within-canopy distribution of incoming solar radiation, so we are not sure if other crops may show the similar phenomena found in our study. I mentioned that Further research is indeed needed to understand the shadedleaf effect on PRI and water stress detection, especially for crops have different canopy geometry from winter wheat.

3. **I don't understand why the authors focus on just PRI. The title only mentions PRI, but the research also includes different forms of NDVI and WI. However, on Page 3, line 7, only different formulations of PRI are mentioned. It seems like a waste of data, when the feature extraction is so limited. There are many other VI's in literature than can be explored, such as; health index (HI), plant senescing reflectance index (PSRI), renormalized difference vegetation index (RDVI), and normalized photochemical reflectance index (PRIn) to name a couple. A nice overview of narrowband indices can be found in (López-López, Calderón, & González-Dugo, 2016).**

We understand review's confusion, so we removed the analysis other than PRI. As reviewed in the introduction, PRI is the physiological indicator that is sensitive to the de-epoxidation state of the xanthophyll pigments, so it has emerged to be a pre-visual indicator of water stress. The other VIs are not physiological index. They are sensitive to the changes in pigment, structure, water content caused by water stress, and thus theoretically less responsive to water stress than PRI.

4. **The terms "stressed plants" and "mixed" are confusing. I assume this is water stressed, but how stressed are they? It would be cleaner to use quantifying terms like RWC. This also applies to figures 3 and 4. It would be interesting to see the trend over RWC and not over days of water stress. Mixed is mentioned several times and shown in figures, but what is it? What percentage of shadow/sunlit?**

Thanks very much for the suggestions. We removed the analysis about the mixed to avoid confusion. And we removed the original figure 3 and 4, and plotted PRI against RWC instead.

5. **A quick google search on "assessing shadow effects on photochemical reflectance index" pointed me at two valuable papers that weren't cited. The authors should consider including them; (Suárez, Zarco-tejada, Sepulcre-cantó, & Pérez-priego, 2008) and (Zhou et al., n.d.).**

Thanks for the suggestion. We had a major revision to the introduction section, and we cited these two studies as suggested.

6. **Page 1, line 1, the authors of this paper look at more than only PRI. The title makes it seem like this is the only Vegetation Index.**

Thanks for the suggestion. We have revised the title and removed the analysis of the other VIs, so this paper focuses only on PRI.

7. **Page 3, line 7, what is the seasonal scale? I don't see this in the rest of the paper.**

We changed it into "in the growth season" to make it clear and consistent.

8. **Page 3, line 30-31, can the authors explain why they use three pots for hyperspectral imaging and three for collecting samples? Why aren't the same pots used for both imaging and sampling? How big is the possible error that is introduced here?**

We separated the pots based on the usage, because we collected samples several times during the experiment, and if we used the same pots for imaging and sampling, the plant density would be different each time after we collected samples. Therefore, to avoid the effect of the decreasing density caused by sampling, we separated the pots into two groups, one group for imaging and one group for sampling. We also averaged measurement taken in three pots to reduce the uncertainty caused by the differences between pots for imaging and pots in sampling.

9. **Page 4, line 1, it is unclear to me what kind of measurements the authors are talking about.**

We specified the measurements as "physiological and spectral measurements".

10. **Page 4, line 16-19, can the authors elaborate a little more on the characteristics of the hyperspectral camera. Mention the FWHM (Full Width Half Max) and the spatial resolution.**

We didn't find the information about FWHM of SOC710, but we calculated the spatial resolution. The revision was highlighted in section 2.3.1.

**11. Page 4, line 21, I don't think you should say "errors", maybe use "variation".**

It has been corrected as suggested. Thanks very much.

**12. Page 4, line 23, can the authors explain more about the wavelength correction? How is this different from the radiometric calibration and what is the in- and output?**

We apologize for this mistake. We checked the user's manual, and found that it should be wavelength calibration (also called spectral calibration) is performed at the factory. Therefore, we removed it from the manuscript.

**13. Page 4, line 27, why do the authors use the Mahalanobis distance method, and what is it? Page 4, line 29, how is the overall accuracy determined? How many validation points are used?**

Since the supervised classification of sunlit and shaded leaves ignored the continuous change of solar radiation received by leaves, we removed the methods of the supervised classification, and manually selected the very dark leaves as shaded leaves and the very bright leaves as the sunlit leaves.

**14. Page 5, line 12-15, the feature extraction in this section is somewhat unclear to me. Can the authors describe why an interpolation function was used and not the closest wavelength to determine the VI's?**

PRI is a normalized difference of reflectance at 531 nm and reflectance at a reference band (e.g. 570 nm) in the visible domain. But the spectral resolution of the hyperspectral imager is about 4.68nm, which means the imager doesn't measure reflectance at 531 and 570 nm. Therefore, to accurately calculate different forms of PRI, we interpolated the original spectrum into 1nm band width.

**15. Page 5, lines 15-19, I'm unfamiliar with this approach of normalizing the difference of a normalized index. It would be helpful if the authors show previous studies that use this approach, or if they could justify this method in another way.**

We removed the analysis about normalizing the difference of PRI_shaded and PRI_sunlit.

16. **Page 6, lines 4-9, the statistical analysis section is very concise and lacks insight in the process. Can the authors make it clear which method is used for which part of the analysis? Further on in the paper the uniform model is introduced, but this is not mentioned in the statistical analysis section. Please include this.**

Thanks very much for your comments. We revised the description of statistical analysis as suggested.

17. **Page 7, figure 2, legend; "sunlit leave". The figure would be more informative if the confidence interval was included (I assume that the data shown is the average response?). How much do the two groups differ?**

Thanks for the suggestion. We added confidence interval in Figure 2.

18. **Page 7, Table 2a&b, this table contains a lot of information, but it is unclear what the most important numbers are. Maybe move this to supplements. Another statistic which can be insightful is a 2-sample t-test and find the VI with the lowest p-value for the most significant difference between sunlit leaves and shadow.**

Table 2a&b show that for PRI570 and similar PRI formulations of sunlit leaves were generally lower than those of shaded leaves for the control treatment, but for the water stress treatment, the reverse relationship was found, which was an important finding of this study. Hence, we decided to keep Table 2a&b in the manuscript.

19. **Page 9, line 12; how is it possible to have a plot of unstressed plants over days of water stress?**

We realized it was confusing, and so we rewrote the sentence.

20. **Page 11, table 3, what do the numbers mean? Is it the Pearson's correlation coefficient? Make this clear in the caption. Page 13, table 4, same comment as for table 3, what is the relationship? Is it a correlation coefficient?**

We specified in the caption that the relationship was the pearson correlation coefficient.

**21. Page 12, figure 5, this figure is not very informative. Either explain why this figure is useful or take it out.**

Thanks for the suggestion. We removed Figure 5.

**22. Page 1, line 26, it states that monitoring the change in canopy temperature is a direct way to evaluate water stress. Shouldn't this be indirect?**

Canopy temperature directly relates to plant transpiration, so it is kind of a direct way to assess water stress. But to avoid confusion, we rewrote the sentence.

**23. Technical Corrections Page 1, line 21, I think it should be "and frequent"; Page 5, line 10, I would leave out the following sentence; "The fraction of shadow and the fraction of sunlit leaves should be summed to 1"; Page 6, line 13, leave out "obviously"; Page 6, line 13, leave out "The spectra of the shadow showed a rise in the near infrared region, but"; Page 8, line 8&10; use "decreased" instead of "fell". Page 14, line1, I think**

We greatly appreciate the thorough review. The above errors have been corrected.

References:

Panigada, C., M. Rossini, M. Meroni, C. Cilia, L. Busetto, S. Amaducci, M. Boschetti, et al. 2014. "Fluorescence, PRI and Canopy Temperature for Water Stress Detection in Cereal Crops." *International Journal of Applied Earth Observation and Geoinformation* 30 (August): 167–78. https://doi.org/10.1016/j.jag.2014.02.002.

Rapaport, Tal, Uri Hochberg, Maxim Shoshany, Arnon Karnieli, and Shimon Rachmilevitch. 2015. "Combining Leaf Physiology, Hyperspectral Imaging and Partial Least Squares-Regression (PLS-R) for Grapevine Water Status Assessment." *ISPRS Journal of Photogrammetry and Remote Sensing* 109 (November): 88–97. https://doi.org/10.1016/j.isprsjprs.2015.09.003.

Rossini, M., F. Fava, S. Cogliati, M. Meroni, A. Marchesi, C. Panigada, C.

Giardino, et al. 2013. "Assessing Canopy PRI from Airborne Imagery to Map Water Stress in Maize." *ISPRS Journal of Photogrammetry and Remote Sensing* 86 (December): 168–77. https://doi.org/10.1016/j.isprsjprs.2013.10.002.

Response to Reviewer#2

Dear reviewer,

We greatly appreciate the reviewers' comments and suggestions, which significantly improve our study. We did a major revision as suggested. Basically, we selected images captured between 12:00-14:00pm as suggested, and thus rerun all the analysis. Minor revisions can be tracked in word document, and major revisions was conducted in several sections highlighted with yellow. Below are the point-by-point responses:

1. **"This study collected PRI data within a window between 1000 and 1400h. Unfortunately, wheat PRI can change dramatically between 1000 and 1400h. Magney et al. (2016) demonstrated that PRI can vary by a factor of 4 between 1000 and 1400h, particularly later in the growing season when water stress is at its peak……"**

   We greatly appreciate the reviewer's comments about PRI data. To minimize the diurnal change in PRI, we selected hyperspectral images capture between 12:00 pm and 1:30 pm. As shown in Magney et al. (2015)'s study, variations in PRI values of wheat canopies were minimal between 12:00 pm and 1:30 pm. The following analyses were based on the images captured between 12:00 pm and 1:30 pm.

2. **"Because foliar deepoxidation state will relate to the instantaneous level of non- photosynthetic quenching (NPQ), and NPQ relates to the instantaneous amount of PAR striking a leaf, what is being defined as a sunlit leaf? In figure 1, some leaves are normal to the camera lens, then curve away. There is more of a continuum of light values, rather than two distinct classes of sunlit canopy vs. shaded canopy……"**

   We do agree that that NPQ relates to the instantaneous amount of PAR striking a leaf, but it's hard to measure PAR received by every leaf. The evaluation of PRI of shaded and sunlit leaves extracted from high-resolution hyperspectral imager is the most effective way to understand the impacts of heterogeneous illumination within a canopy on PRI as well as water stress detection with PRI. However, we realized the problem of classifying the whole hyperspectral images into shaded and sunlit leaves. Therefore, we

manually selected the most shaded and the most sunlit leaves of each image, as conducted in several similar studies (Takala and Mõttus 2016; Zhou et al. 2017). The following analysis were all based on the data of the manually-selected shaded and sunlit leaves.

3. **The variants of PRI selected for this study are influenced by both long term (constitutive) and short term (facultative) plant physiological processes, and the influences of long term vs. short term pigment pools cannot be isolated from each other.**

   Hwang et al. (2017) found that the ratio (sPRI) of sunlit canopy PRI (backward direction images) to shaded canopy PRI (forward direction images) captured drought signals in a temperate deciduous forest. Inspired by these studies, we tried to use ΔPRI and PRI_sunlit/PRI_shaded to disentangle the facultative and constitutive components. However, the correlation between RWC and ΔPRI or PRI_sunlit/PRI_shaded was not significant, and thus results were not shown in the manuscript.

   We PRI3 that used reflectance at 512 nm as the reference band provided the most accurate estimates of RWC with varying shaded-leaf fractions, except for the 100% shaded-leaf fraction. PRI3 was originally developed for the needle tree based on the evidence that reflectance at 512 nm was not responsive to the change in xanthophyll epoxidation state (Hernández-Clemente et al. 2011). In their study, PRI3 showed the highest correlation with the stomatal conductance and water potential at the canopy level, and the lowest sensitivity to canopy structure, in comparison with PRI570 and NDVI. Our results also showed the superior performance of PRI3 than the other formulations of PRI in estimating RWC, implying that for winter wheat band 512 nm might be a better reference band that could maximize the physiological responses of band 531 nm.

4. **The light (PAR) incident on the plants was not measured or considered in the analyses. This, combined with the fact that the authors are treating a continuous variable of light intensity as a categorical variable (i.e. sun- lit vs. shaded) unfortunately are fundamental omissions that make the results of this study invalid in my opinion.**

   Thanks very much for the comments. Although we didn't measure PAR at the site, the reference panel was captured in every image for the radiometric

calibration. As the reference panel is spectrally neutral across a wide range of wavelengths, we averaged the DN values within 450- 650nm as the surrogate of PAR. However, we found that the correlation between PRI and PAR was not significant, except for PRI5, and thus results were not shown in the manuscript (Figure 1).

Furthermore, our study focuses on quantifying the impacts of varying shaded-leaf fractions on water stress detection using PRI. The sunlit and shaded leaves represent the within-canopy distribution of incoming solar radiation, because it is hard to measure solar radiation received by every leaf. Therefore, from our perspective, the investigation of how PRI_sunlit and PRI_shaded respond to the changing solar radiation under the control and water stress condition is not critical to our study.

[Figure]

Figure 1. Pearson correlation coefficient between PRI and DN value of reference spectral panel of unstressed plants and stressed plants for sunlit leaves (a) and shaded leaves (b). Correlation coefficient higher than 0.515 is significant at P < 0.05.

5. **Please remove "the" in the title to make the title more readable.**

   We removed it as suggested. Thanks!

6. **Abstract Line 7: PRI can correlate with several types of plant ecophysiological functions depending on the timeframe of analysis. As a result, several different types of plant stress (nutrient, water, pathogen) can influence the de-epoxidation state of a plant as it responds to excess light. It would also be helpful to indicate here that the PRI is a remotely sensed spectral vegetation index. I therefore suggest that the authors remove 'water' from this first line and rather state ". . .a pre-visual remotely sensed indicator of plant stress."**

   We revised abstract as suggested.

7. **Line 8: ". . .whether variations the shadow fraction, which can be influenced by varying view angle and crop density. . ."**

   We rewrote the sentence.

8. **Line 11: Three different variants of PRI are presented here, without any indication that the different formulations of PRI can be interpreted in different ways, based on several considerations (e.g. timeframe of analysis, phenological responses within the growth cycle, etc.). Specifying how many formulations were tested would be helpful. For example, ". . .using 6 different formulations. . ."**

   Thanks for the suggestion. We specified the number of formulations of PRI we used.

9. **Introduction Page 1, Line 23: Suggest new paragraph starting with "Remote sensing. . ." Line 24: ". . .assess water status. . ."**

   Thanks for the suggestions. We have revised as suggested.

10. **Page 2, line 4: Please see first comment I made on the abstract. As I read further in this section, it appears that the authors do a pretty good job of overviewing the connection between PRI and stress. It would be useful to still include some mention of non-water-related stressors here in this section (e.g. nutrient availability, pathogens) and also structure the argument to explain to the reader that the xanthophyll cycle serves to**

**protect the light harvesting complex from excess light, and that the threshold for a plant to deal with excess light varies according to a multitude of environmental factors (water availability being one, yes, but not the only one).**

Thanks for the suggestion. We revised introduction section, and mentioned environmental stress (not just water stress) can trigger the de-epoxidation.

11. **Line 25: I respectfully disagree that the impacts of shadowing on PRI are generally ignored. This is a widely recognized phenomenon. The authors should re-word this statement.**

The sentence has been deleted.

12. **Line 30: The more mechanistically appropriate way to explain this is that the xanthophyll cycle status is affected by incident PAR, which is in turn affected by the level of self-shading (i.e. shadow fraction) within a canopy.**

The sentence has been revised as suggested.

13. **Intro in general: Magney et al. (Remote Sensing of Environment, 2016, 173: 84-97) reported relationships between derivations of the PRI and various environmental conditions (including VPD) in three different portions of a wheat field. This paper should be briefly overviewed in the introduction and potentially the discussion due to the goals of the study being closely related to the current paper.**

Thanks for the suggestion. Magney et al. 's study (2016) was mentioned in the introductions and discussed in the discussion section.

14. **Page 3, Line 2: ". . .the PRI's capability. . ." The PRI does not have agency or capability. Scientists have a capability to interpret PRI data to detect stress. Please reword. Materials and Methods Page 3, Line 13: "rectangular" Page 3, Line 16: "plots" should be "pots" Line 20: remove "a"**

Thanks very much for the thorough review. The minor errors were corrected.

15. **Line 22: Were all pots located in the rainout shelter? Because plant light harvesting complexes and pigments can change as a result of the**

**ambient light environment in which they are growing, it is important to know whether the rain-out shelters reduced the PAR striking the canopy. What was the influence of these shelters on the ambient light condition?**

Yes, all pots located in the rainout shelter. The rainout shelter had transparent curtains, which were rolled up on the sunny days. And as we mentioned in the manuscript, the transparent shed was open half an hour before measuring began. Although we didn't compare the light environment for the pots inside the rainout shelter and outside the rainout shelter, we believe the differences would be minimal for the seasonal water stress detection.

16. **Line 23 and 26: "control" is more specific than "reference"**

    It has been corrected as suggested.

17. **Line 20: Due to a number of factors such as VPD and air temperature, the PRI of wheat can change dramatically between 1000 and 1400h……**

    Please see the response to the first comment.

18. **Line 30: Depending on the specs of this journal, you probably need to specify the ENVI manufacturer, version, etc.**

    The manufacturer and version were added as suggested.

19. **Figure 1: This classified image highlights a question for me. Because the deepoxidation state will relate to the instantaneous level of NPQ, and NPQ relates to the instantaneous amount of PAR striking a leaf, what is being defined as a sunlit leaf? In figure 1, some leaves are normal to the camera lens, then curve away. There is more of a continuum of light values, than two distinct classes of sunlit canopy vs. shaded canopy. As a result, the analysis may be flawed because it is trying to capture a process that responds to a continuum (of incident PAR, specifically) using a binary shadow/non-shadow classification.**

    Please refer to response to comment #2.

20. **Table 1: Unfortunately, of all of the PRI calculations used in this study, the two variants of PRI that have been shown to correlate most strongly**

**with water status and other diurnally changing physiological variables, the deltaPRI and the PRIo, (Magney et al. 2016) were not calculated or used in this study. The various variants of PRI used in this particular study are influenced strongly by longer-term chlorophyll:carotenoid ratios that will mask the instantaneous effects of changing light or VPD conditions**

Please refer to response to comment #3.

**References:**

Hernández-Clemente, Rocío, Rafael M. Navarro-Cerrillo, Lola Suárez, Fermín Morales, and Pablo J. Zarco-Tejada. 2011. "Assessing Structural Effects on PRI for Stress Detection in Conifer Forests." *Remote Sensing of Environment* 115 (9): 2360–75. https://doi.org/10.1016/j.rse.2011.04.036.

Hwang, Taehee, Hamed Gholizadeh, Daniel A. Sims, Kimberly A. Novick, Edward R. Brzostek, Richard P. Phillips, Daniel T. Roman, Scott M. Robeson, and Abdullah F. Rahman. 2017. "Capturing Species-Level Drought Responses in a Temperate Deciduous Forest Using Ratios of Photochemical Reflectance Indices between Sunlit and Shaded Canopies." *Remote Sensing of Environment* 199 (September): 350–59. https://doi.org/10.1016/j.rse.2017.07.033.

Takala, Tuure L.H., and Matti Mõttus. 2016. "Spatial Variation of Canopy PRI with Shadow Fraction Caused by Leaf-Level Irradiation Conditions." *Remote Sensing of Environment* 182 (September): 99–112. https://doi.org/10.1016/j.rse.2016.04.028.

Zhou, Kai, Xinqiang Deng, Xia Yao, Yongchao Tian, Weixing Cao, Yan Zhu, Susan Ustin, and Tao Cheng. 2017. "Assessing the Spectral Properties of Sunlit and Shaded Components in Rice Canopies with Near-Ground Imaging Spectroscopy Data." *Sensors* 17 (3): 578. https://doi.org/10.3390/s17030578.

---

## Author Response (AR2)

**Responses to Reviewer**

1. For the analysis on the slope and intercept the authors have chosen to use a linear regression. However, when you look at figure 5 it is clear that for example an exponential fit could be a better representation. The model misses the point of 100% shaded, for 3 out of 4 subfigures). A quick pseudo data analysis led me to the following equation: PRI3_intercept=a*exp(shaded fraction * b)-c, where a=9.1e-7, b=13.37, and c=0.31, this results in an R-squared value of 0.997. Why is it that the authors decide to use a linear model?

Since the data had negative values, a quadratic function, instead of a linear model, was applied to describe the relationship between the slope/intercept and RPI.

2. IF, the authors decide to use above mentioned model the conclusion of the paper that "PRI of the pure shaded leaves may yield inaccurate estimates of plant water status" could be more refined. I do understand that when they use the linear models for estimating RWC using PRI the authors chose to use the regression from the 50/50 ratio sun-shaded leaves. However, if an estimation of the true shaded area can be made, the model geared towards that specific ratio could be used for RWC estimation.

It would be interesting to quantify the impact of using a particular linear model for the estimation of RWC using all different models (trained on the different rations). This sensitivity analysis might show whether the decision of the authors to used 50/50 leaves/shadow ratio for RWC estimation is acceptable. How does the RMSE look when 100% sunlit-model is used? And how far off is the estimation from the 'best' estimation (probably reached with the model that it's trained on).

Thanks very much for the valuable suggestion. We added Figure 6 to show RMSE of RWC estimated with the linear regression model derived from the PRI of the sunlit leaves/shadow ratio of 50/50 and with the linear regression models geared towards the known shaded-leaf fractions.

4. I also strongly recommend that a native English speaker thoroughly examines the manuscript for language errors. I've pointed out a couple, but I'm sure that I've overlooked others.

Thanks very much for your suggestion. The manuscript was edited by a native English speaker.

5. Page 6, section 3.1: There is no reference in the text to Figure 3 and Table 2.

Thanks very much for the suggestion. The references to Figure 3 and Table 2 were provided and highlighted in section 3.1.

6. Page 8, figure 3: The authors should consider using different symbols for the sunlit and shaded leaves data points. There is also no mention what the error bars mean. Is it for example a standard deviation, or a quantile? I would also like to know what the authors think causes the peak in shaded leaves at RWC 0.5-0.6.

We revised Figure 3 as suggested. We hypothesized that the peak in shaded leaves at RWC 0.5-0.6 was the combined effect of the uncertainty in spectra measurements and the optimal RWC that maximized the photosynthetic rate in the shaded leaves. However, we haven't found any reference to support our hypothesis. We would greatly appreciate if the reviewer could provide any suggestions.

7. Page 8, line 16: The terms "strong" and "weak" seem somewhat ambiguous. I don't see how a R-squared of 0.31 is strong, and a R-squared of 0.28 is weak. Figure 4 a) does not show a strong relationship either.

We rewrote the whole sentence to make it clearer.

8. Page 9, figure 4: I would like to compare this figure with the data in table 3. However, the authors use a different statistical metric. They should consider changing from Pearson's r to R-squared; Page 10, line 6: I assume the authors are showing Pearson's r in table 4. But it would be appreciated if they explicitly say this. Again, reconsider changing this to R-squared so it matches figure 5.

We changed the Pearson's r to R-square as suggested.

Technical Corrections

Page 3, line 12: "water stress experiments of" ◊ "a water stress experiment on"

Page 3, line 22: "the volumetric" ◊ "a volumetric"

Page 3, line 22: "at the field" ◊ "at field"

Page 3, line 26: "were growing outdoor under the natural condition" ◊ were grown outdoors under natural conditions"

Page 3, line 28: "of the field capacity" ◊ "off field capacity"

Page 3, line 28: "The water stress" ◊ "Water stress"

Page 3, line 29: "which was the tiller" ◊ "which was during tiller"

Page 3, line 29: "prevent the external" ◊ "prevent external"

Page 5, line 1: "the radiometric" ◊ "radiometric"

Page 5, line 10-11: "in the MATLAB software" ◊ "in MATLAB".

Page 10, line 3&4: make the "P" lower case.

Page 13, line 27: "proved" ◊ "proven"

Page 13, line 27: "of the structural" ◊ "of structural"

Page 14, line 30: "for crops have different" ◊ "for crops that have a different"

We greatly appreciate the reviewer's thorough editing. The corrections have been made as suggested.

[revised manuscript text omitted]